# INKORRECT: DIGITAL INK SPELLING CORRECTION

**Andrii Maksai, Henry A. Rowley, Jesse Berent & Claudiu Musat**
Google Research
{amaksai, har, jberent, cmusat}@google.com

## ABSTRACT

We introduce Inkorrect, a digital ink (online handwriting) spelling correction approach. We show that existing metrics don't capture the quality of spelling correction, and propose a new one. Our approach outperforms previous work in automated and human evaluation, while also being more data- and label-efficient.

## 1 INTRODUCTION

Digital ink offers new experiences for drawing and full-page note-taking, ex. with a stylus on a tablet. Correcting errors using pen and paper involves finding the error, deleting affected strokes, and replacing them with new strokes. With Digital Ink Spelling Correction (DISC), errors can be corrected in one go. However, getting human-annotated data to train a model for DISC is difficult and costly. Collection requires multiple users to cover writing style variability, rewriting their own (corrected) samples, and adjusting the positions of multiple words to accommodate the correction.

Existing DISC methods thus resorted to style transfer as a proxy for spelling correction. These still require cumbersome data collection (multiple samples from the same writer or character segmentation) and yet fail to capture high-frequency details of how a particular person wrote a particular sample. Evaluating the quality of generative models is also tricky, and most approaches resort to human evaluation, which doesn't scale.

We propose a method for doing DISC that does not require character segmentation or multiple samples from the same writer. We propose measuring DISC quality on two axes, recognizability and newly proposed similarity metric, and describe a recipe for automatic comparison of two models that correlates well with human evaluation. Our method is preferred to previous work by all metrics.

## 2 RELATED WORK

**DISC.** Digital ink synthesis work ranges from the LSTM-based work of Graves (2013), to Transformers (Ribeiro et al., 2020) and GAN-based models (Song et al., 2018). DISC, however is a new endeavor and comparatively little has been published on it. Style is the central element in the only work addressing DISC, DeepWriting (Aksan et al., 2018). Style transfer has been addressed by either priming the model on an input (Graves, 2013), or by learning a style embedding, either global, or per-character (Kotani et al., 2020; Aksan et al., 2018; Chang et al., 2021).In DeepWriting (Aksan et al., 2018), the authors use a VRNN (Chung et al., 2015) as the backbone for synthesis, and learn

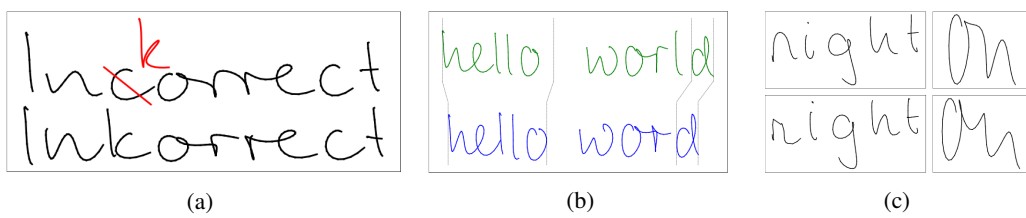

(a)         (b)         (c)

Figure 1: **(a)** High fidelity ink spelling correction with our approach. **(b)** Alignment of parts of the original and spell-corrected ink found by the proposed $CDE$ metric. **(c)** Examples of recognizability-similarity trade-off in DISC. Very high similarity hurts recognizability: is it "night" or "right", "on" or "an"?

a per-letter style representation. This allows spelling correction in which letters are highly similar to the original writing, but the approach requires character segmentation of the input, which can be error-prone. As the authors note, the method performs poorly on samples with delayed strokes and cursive writing, likely because it assumes monotonic segmentation.

**Evaluation.** Digital ink synthesis approaches typically use following metrics for evaluation: **(i)** human evaluation (Ribeiro et al., 2020; Das et al., 2021), **(ii)** recognizability using a separately trained recognizer/classifier (Cao et al., 2019; Chang et al., 2021), and **(iii)** similarity, typically to measure reconstruction error, using Chamfer distance, Sliced-Wasserstein distance, or Fréchet Inception Distance (FID) (Song, 2020; Aksan et al., 2020; Das et al., 2021). In Section 3 we outline the shortcomings of these metrics for DISC and propose an alternative.

## 3 EVALUATION

While creating spell-corrected labels is trivial, creating high-fidelity paired samples of original and spell-corrected inks is difficult. We therefore introduce a method to evaluate the quality of handwriting spelling correction without access to ground truth spell-corrected inks. We assume that, as in DeepWriting (hereafter DW), the generative model that takes an original ink and a spell-corrected label, and produces a spell-corrected ink. We describe recognizability and similarity metrics, and show how in DISC there exists a trade-off between the two, in Figure 1c.

**Recognizability** We use a separately trained recognizer to evaluate how recognizable the synthetic ink is. Similar to Chang et al. (2021), we use Character/Word Error Rate ($CER$/$WER$) metrics.

**Similarity** between a real and synthetic ink can be measured either in the input space (ex. Chamfer Distance ($CD$)) or in the feature space (ex. FID). In DISC however, we don't want a perfect reconstruction – rather, most of the ink needs to be reconstructed, except for the parts that have been spell-corrected. We propose a relaxation of $CD$ (which computes pairwise minimum distances between two points clouds) to better capture the distance between original and spell-corrected ink.

Below we use $P = \{p_i \in \mathbb{R}^2 | 1 \leq i \leq |P|\}$ and $Q$ to identify the points of the original and spell-corrected inks, ordered by *the x-coordinates of the points*. For inks, written nearly horizontally in left-to-right languages, this means earlier points roughly correspond to earlier letters.

**Chamfer Distance Edit-aware** ($CDE$), our proposed metric, splits inks into $K$ groups $(P_1, \ldots, P_k)$, $(Q_1, \ldots, Q_k)$ and aligns different groups independently to account for shifts and changes introduced by the spelling correction. $CDE$ is a relaxation of $CD$, and is equivalent to it for $K = 1$. It is computed as $CDE(P, Q, K) = \min_{(P_1, \ldots, P_k), (Q_1, \ldots, Q_k)} \sum_{i=1}^{K} CD(P_i, Q_i)$.

To account for misalignments introduced by the correction, we select $K$ to be number of words in the original label plus edit distance between original and spell-corrected label. $CDE$ can be optimized via dynamic programming to compute $F_{|P|,|Q|}^K$ where $F_{i,j}^k$ is the distance when grouping the first $i$ points of $P$ and first $j$ points of $Q$ into $k$ groups to minimize a sum of $CD$:

$$F_{i,j}^k = \min_{l < i, m < j} F_{l,m}^{k-1} + CD(\{p_l, \ldots, p_i\}, \{q_m, \ldots, q_j\})$$

**Limitations** $CDE$ assumes nearly horizontally written inks of the same scale (which is the case for datasets we used). In general, external line height / writing angle estimation models may be needed.

**Human evaluation protocol** We presented participants with a triplet of samples (original ink, spell-corrected ink from Inkorrect, spell-corrected ink from DW) and asked which correction they preferred. This helped us answer three questions: **(i)** Does the recognizability metric correlate with human preference? **(ii)** Which similarity metric correlates best with human preference? **(iii)** Is our DISC approach preferred to the existing baseline? Participants were also asked to reflect on the criteria they used for selecting the best spelling correction. We present the results in Section 5.

## 4 METHOD

Our generative model, shown in Figure 2 (left), is a multi-layer LSTM with monotonic attention over the label, similar to Graves (2013); Chang et al. (2021). The key difference is the style extraction

Figure 2: Left: Architecture. Right: Effect of similarity masking and comparison with DW. **(a)** Original ink; **(b)** *sim* 1.0; **(c)** *sim* 0.4; **(d)** *sim* 0.0. **(e)** DW. Similarity gradually decreases as we increase masking of the style.

block which finds a latent representation of the original ink passed as input to the model. Empirically it captures both global features like angle and size, and local features like the individual letter shapes.

**Tradeoff Control through Masking** The main novelty is a feature masking layer that can mask parts of the style vector. During training, the model takes the original ink as input, extracts its style, and uses it together with the original label to reconstruct the original ink in a teacher-forced manner, minimizing the negative log-likelihood. To stop the model from memorizing the original ink through the style layer, for each training sample we mask each feature of the style vector with a masking probability, chosen for each sample uniformly at random between 0 and 1.

**At inference time** the model takes as input the original ink and the spell-corrected label, as well as a *similarity* (*sim*) value, which controls the amount of information that will be allowed to flow through the style. Given a *sim* value of $X\%$, we mask the last $100 - X\%$ of the features in the style vector. See example and comparison to DW in Figure 2.

**Implementation details** We select the best model as the one with the lowest $CER$ on the validation data when using a *sim* value of 1. Directly maximizing NLL instead leads to overfitting. We use Adam (Kingma & Ba, 2015) with learning rate 0.001 and a batch size of 256.

**Limitations** A fixed style dimension does not scale to very long inputs and could be aided by variable length style (Skerry-Ryan et al., 2018) or by reinstating character segmentation.

## 5 RESULTS

**Datasets** We use the **HANDS-VNOnDB** (Nguyen et al., 2018) and **DeepWriting** (Aksan et al., 2018) datasets. They exhibit varied writing styles, input lengths, stroke orders, and come from different languages. Since they feature only ink-label pairs, we augment them with spell-corrected versions of the labels.

**Spell-corrected label generation** Spelling correction replaces a word with another known (ie dictionary) word. Based on a list of the most common English misspellings, most spelling corrections (71%) have an edit distance of 1 to the original word, and most of the rest have edit distance of 2. We use this information to generate the spell-corrected labels: for each sample, we pick one word and replace it with a random dictionary word at an edit distance of 1 or 2 with probability 71/29%.

### 5.1 HUMAN EVALUATION

Evaluation involved side-by-side comparison with DW on the **DeepWriting** test set, with randomized image order. Each sample was seen by 3 of the 10 participants. In open-ended feedback, 9 of 10 participants stated that the first thing they looked at was the recognizability (could they parse the spelling corrected ink as the intended label), and if the answer was yes for both samples, they started looking at the similarity traits (letter size, angle, cursiveness, shape of individual letters, etc).

**Recognizability-preference correlation** When one of two inks is correctly recognized by the recognizer, humans prefer it (73% of the cases). If the recognized sample is from Inkorrect, it is preferred in 90% of the cases, and in 62% of the cases if it is from DW. This gap can be explained by additional potentially desirable traits, such as a high degree of smoothness. Overall, this underscores the relevance of recognizability in the evaluation of DISC.

**Similarity-preference correlation** Since we've established that people prefer samples with higher similarity **iff** they can recognize both inks, all evaluations we performed in this section have been limited to samples where outputs of both models can be recognized by the recognizer.

Figure 3: **Left, top:** Examples of spelling correction, **HANDS-VNOnDB** dataset, *sim*=1.0. Note how correcting "v" to "x" with too much similarity hurts recognizability in the third column. **Left, bottom:** How often is the sample with lower distance metric preferred in human evaluation, depending on the difference in distance metric. **Right:** $CER$ and $CDE$ metrics for Inkorrect and DW.

Since similarity is a continuous, rather than a binary value, for slight differences, the amount of noise can overwhelm the signal. As in Musat et al. (2011), we emphasize that, as the differences between the two models get larger, lower $CDE$ almost always correlates with human preference.

We rank all samples by the absolute difference in distance metric $|D_{\text{Inkcorrect}} - D_{\text{DW}}|$ ($D$ is the similarity metric, either $CD$ or $CDE$, between original and sample synthesized by the two methods). We threshold this difference and show that as the threshold grows, the samples with the lower distances are preferred by humans more frequently. As shown by the Figure 3 (left, bottom), lower values of $CDE$ agree with human preferences up to 85%, while it is only up to 66% for $CD$. This underscores that $CDE$ is closer to human evaluation than previously used $CD$.

**Comparison to DW** Our model is preferred in $66\% \pm 4.7\%$ of the cases. Our proposed automated evaluation, comparing samples based on recognizability-then-similarity, agrees with human evaluation in 79.4% of the cases, further validating agreement between automated metrics and human judgement and providing a way for practitioner to compare two models. Examples in Figure 2, right.

## 5.2 AUTOMATED EVALUATION METRICS

After validating the recognizability and similarity metrics, in this section we used them study the performance of the proposed method and compare it to the state of the art. We further show that by varying the $sim$ value, we can trade-off between the similarity ($CDE$) and recognizability ($CER$/$WER$).

Since DW does not present a way to vary the amount of information in the style embedding, we report only two points: $sim = 1.0$ (extracting the style from the original ink) and $sim = 0.0$ (model not conditioned on any input). For our model, we use *sim* values between 0 and 1 in 0.2 increments.

As seen in Figure 3 (right), for maximum similarity, Inkorrect has better $CDE$ and $WER$ with approximately similar $CER$. When 20% of features are masked, Inkorrect is better on all axes than DW conditioned on the original ink, and without any style information, Inkorrect is preferred to the version of DW not conditioned on original ink. We hypothesize that Inkorrect's lower error rates are due our model not being conditioned on potentially incorrect segmentation information.

The **metrics trade-off** is clearly visible in Figure 3 (right). A model trained without feature masking yields even better similarity but harder to recognize inks ($CDE$=4.3, $CER$=12.9), giving another example of the trade-off, and underscoring the importance of the feature masking: without it, the content of the original ink leaks into the style embedding.

## 6 CONCLUSION

We proposed Inkorrect, the first method that balances recognizability and similarity in digital ink spelling correction. It is data-efficient and privacy-enabling, as it does not need multiple samples from the same writer, and resource-efficient, removing dependencies on character segmenters.

We study the tradeoff between various recognizability and similarity measures and introduce a DISC-specific metric ($CDE$), that accommodates the expected differences coming from correcting the spelling. We show these correlate well with human judgment and use them to evaluate Inkorrect. Its Pareto frontier dominates the prior work.

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
