# OpenReview forum: "Inkorrect: Digital Ink Spelling Correction"
_ICLR.cc/2022/Workshop/DGM4HSD — ICLR 2022 DGM4HSD workshop Poster_

### Official Review · Reviewer_BBkD · 2022-03-14
**work of somewhat novel ideas, but with poor presentation**

**Rating:** 6
**Confidence:** 2

**Review:**

The paper described a new method to auto-correct handwritten words.

Pros:
1. The paper provides a new way of evaluating the performance of the spell-correcting model. It argues that recognizability, often used as an evaluation for models in this field, is not sufficient. A new metric to take similarity between original and corrected text should be taken into consideration as well. The authors also give a math formula for the new metric and discuss its limitation.
2. The paper proposes to use masking and reconstruction techniques to learn the representation of the original ink, which is claimed to be novel.
3.  For a visual application, it's good to see human participants involved in the evaluation to assess how well the model recovers the spelling errors and how much fidelity it is to the original ink.

Cons:
1. The paper is hard to follow because the architecture of the model is not summarized in the abstract, conclusion, overview figure. So it's very difficult to get a general idea of how this model works without reading the whole paper. Same with other figures and citations, they are supposed to be standalone, giving a general idea of how the model works. But it is not the case with this paper.
2. The authors fail to elucidate the impact this handwriting correction real life. It seems to me that most texts nowadays are processed digitally. Even hand-written texts can use optical character recognition (OCR) to convert into their digital form. Any spelling problems can then be corrected digitally, which is of a very high success rate. So I don't know the use scenario of this work. The impact should be explicitly told to the readers.
3. Lack of enough explanations for concepts. For example, the concept of the trade-off between recognizability and similarity is not well explained in texts. Same the related figure 1(c) that aims to explain this trade-off. It has no annotations so it's not clear what the top and bottom sub-figures mean. They look the same to readers. Same with monotonic attention. What is this?

In conclusion, even though the paper lacks a clear presentation for its method, there are some merits in its method and evaluation. Hence a weak accept is granted. More work should be done to provide the readers with ease of reading.

---

### Official Review · Reviewer_mFY4 · 2022-03-17
**An interesting addition to the domain literature and appropriate for the workshop**

**Rating:** 7
**Confidence:** 3

**Review:**

The paper proposes an architecture for digital ink spelling correction i.e., finding and correcting errors in handwritten digital text. I believe that the spirit of the paper is in line with the aims of the workshop---it proposes a generative model for a structure data type that is not a typical benchmark in machine learning tasks.

The main technical novelty of the paper compared to previous approaches is the addition of a random masking layer in the style encoder, which regularizes the effect of writing style on the downstream prediction of corrected text. The amount of regularization can be controlled at test/validation time, allowing for a trade-off between writing style similarity and optimal correction.

Although the remainder of the architecture is in large part derived from existing literature, I believe that the new additions are interesting and, coupled with the use of new evaluation metrics, would make this paper a good addition to the workshop. The paper is written and presented well-enough given the limitations associated with the workshop format (in particular the main ideas are summarized well in the first two figures), although some clarification aimed at those who are not domain experts would have been helpful (e.g., defining the character/word-error rate for those not in the field).

---

### Decision · Program_Chairs · 2022-03-25

Accept (Poster)